# Desymmetric homologating annulation to access chiral pentafulvenes and their application in bioimaging

Sanjay Singh[1], Ravi Saini[1], Akshay Joshi[2], Neetu Singh[2] & Ravi P. Singh[1] ✉

The architectural design of polycyclic/multisubstituted pentafulvenes has demonstrated great potential for the development of electrochromic materials and biologically active motifs. Unfortunately, the enantioselective construction of such distinctive cores with all carbon quaternary chiral centers has remained untouched to date. Herein, we disclose an enantioselective homologating annulation of cyclopent-4-ene-dione with 3-cyano-4-methylcoumarins through L-tert-leucine derived thiourea catalysis, affording a wide range of enantioenriched polycyclic multisubstituted embedded aminopentafulvenes with excellent stereocontrol (up to 99:1 er) and chemical yields up to 87%. A detailed photophysical and cytotoxicity analysis of racemic and chiral homologated adducts unveils the exceptional behavior of chiral adducts over their racemic analogs, highlighting the importance of stereoselectivity of the developed scaffolds. A cellular uptake experiment in a mammalian fibroblast cell line confirmed the potential of developed polycyclic aminopentafulvene cores as a highly promising labeling dye that can be utilized for bioimaging without any adverse effects.

Fulvenes represent a highly fascinating class of organic frameworks that constitutes a distinct π-electron system due to their cyclic cross-conjugation[1]. Owing to the size of the ring system, fulvenes have been classified either as tria, penta, hepta, or nonafulvenes[2,3]. In particular pentafulvene, a cyclic benzene isomer exhibiting high polarizability and non-benzenoid aromaticity initially attracted chemists to pursue research in this field[4,5]. The organic scaffolds containing fulvene core are frequently encountered in biologically relevant motifs[6–8], electrochromic dyes[9], and solar cells[10] and also demonstrate exciting physical and chemical properties (Fig. 1A)[11,12]. Considering its importance, a plethora of synthetic procedures to access pentafulvene and its derivatives have been developed over the years[12,13]. However, the stereogenic synthesis of chiral pentafulvenes/embedded polycyclic pentafulvenes has remained untouched to date.

In an expedition to synthesize polycyclic arenes[14], we were primarily fascinated by the work of Badiya et al. based on Brønsted base mediated vinylogous (4 + 2) annulation between alkylidene malononitrile and cyclopent-4-ene-1,3-dione[15]. This led us to envision an organocatalytic vinylogous desymmetrizing (4 + 2) annulation between 3-cyano-4-methylcoumarin and cyclopent-4-ene-1,3-dione to construct chiral polycyclic coumarins containing highly substituted aniline motif, with an idea that the methyl and the cyano groups bearing coumarin will act as donor and acceptor ends, respectively, to facilitate the desired transformation (Fig. 1D). To our surprise, we did not obtain any (4 + 2) annulated product, instead, a unique desymmterizing homologation of cyclocpent-4-ene-1,3-dione, and subsequent annulation provided multi-substituted polycyclic aminopentafulvene core bearing remote all-carbon quaternary stereocenter (Fig. 1E).

Having realized this serendipitous result, a thorough search of the literature revealed that such an organocatalytic desymmetrizing homologation of cyclic enones and subsequent annulation strategy has never been utilized for the enantioselective synthesis of centrally chiral polycyclic embedded amino pentafulvenes. However, traditionally, some strategies that have been successfully explored for

[1]Department of Chemistry, Indian Institute of Technology, Delhi, Hauz Khas, New Delhi 110016, India. [2]Center for Biomedical Engineering, Indian Institute of Technology Delhi, Hauz Khas, New Delhi 110016, India. ✉e-mail: ravips@chemistry.iitd.ac.in

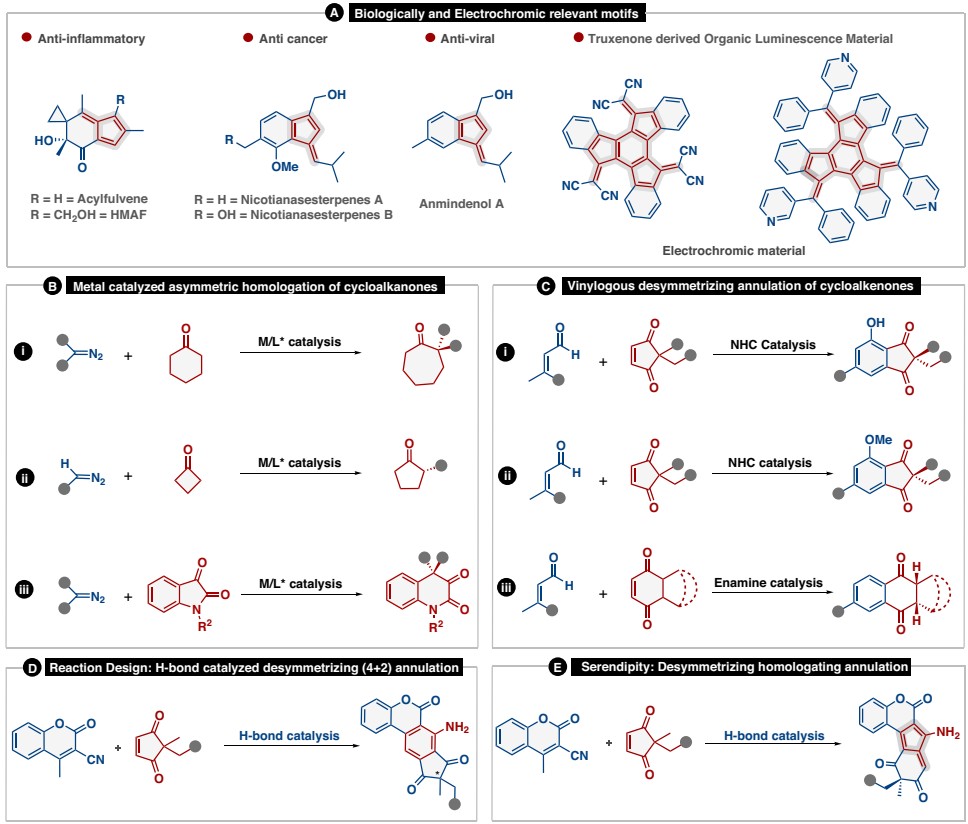

**Fig. 1 | Reaction background and our discovery. A** Biologically relevant and electrochromic motifs containing pentafulvenes. **B** Metal catalyzed enantioselective homologation of cycloalkanones. **C** Asymmetric vinylogous desymmetrizing annulation of cycloalkanones. **D** Reaction design for H-bond catalyzed desymmetrizing (4 + 2) annulation. **E** Asymmetric vinylogous enantioselective homologating annulation between cyclopent-4-ene-1,3-dienone and 3-cyano-4-methylcoumarin.

homologation of cyclic ketones by C-C insertion are the direct addition of diazo alkanes[16], β-oxido carbenoids[17], α-lithoalkyl sulfones[18], and N-substituted benzotriazoles/bis-benzotriazole derivatives[19] to reactive carbonyl compounds. On the other hand, enantioselective homologation using ketone substrate has largely been focused on either symmetrical cyclic ketones or reactive carbonyl compounds such as isatin. In this regard, Maruoka et al. have reported a chiral aluminum Lewis acid-catalyzed desymmetrizing asymmetric ring expansion of cyclohexanones with α-diazoacetate (Fig. 1B (i))[20]. In another report, Kingsbury et al. realized a highly enantioselective desymmetrization with diazomethane utilizing a chiral Sc(III)-trisox catalyst (Fig. 1B (ii))[21]. Recently, Feng et al. have described an enantioselective intramolecular homologation of isatins with α-diazoesters by using chiral Sc(III)-N, N′-dioxide complex (Fig. 1B (iii))[22]. Further literature investigation revealed that there are very few reports on desymmetrizing enantioselective (4 + 2) annulation reactions involving cyclic enones (Fig. 1C)[23–25]. However, an enantioselective homologating annulation is still to be realized.

Here in this article, we report a groundbreaking discovery by synthesizing multisubstituted polycyclic chiral amino pentafulvenes through catalytic desymmetrizing homologated annulation between 3-cyano-4-methylcoumarin and cyclopent-4-ene-1,3-dione.

## Results and Discussion

Our exploratory studies for desymmetrizing homologation reaction commenced by taking 3-cyano-4-methylcoumarin (**1a**) and prochiral cyclopent-4-ene-1,3-dione (**2a**) as model substrates, **C1**, a quinine based thiourea as catalyst, O₂ as an oxidant, and DCM as reaction medium (Table 1). Pleasingly, when the catalytic experiment was carried out at rt for 3 d, the desired homologated product **3a** was obtained in 44% yield with poor stereocontrol (Table 1, entry 1). A variety of

other organocatalysts (**C2-C6**) were evaluated to further enhance the reactivity and selectivity (Table 1, entries 2-6). To our delight, **C6**, an L-*tert*-leucine-based dipeptide thiourea turned out to be the most promising catalyst by providing the homologated product **3a** in 50% yield and high stereocontrol (Table 1, entry 6). To further improve the selectivity, various other solvents were screened (See ESI for details). Among all the screened solvents, CHCl₃ proved to be the best reaction medium in terms of reactivity and selectivity (Table 1, entry 7) (Supplementary Table 1-3 for detailed optimization). Gratifyingly, when the reaction was performed at 0 °C, **3a** was obtained in 73% yield with an excellent enantiomeric ratio of 97:3 (entry 10).

With the optimum reaction conditions established, we first evaluated the generality of our protocol by reacting a wide range of 3-cyano-4-methylcoumarins (**1a-1t**) with cyclopent-4-ene-1,3-dione bearing all carbon prochiral center **2a** (Fig. 2). The coumarins, regardless of electronically and sterically modified substituents at different positions on the *benzo*-core were very well tolerated and underwent desymmetrizing homologation to yield the corresponding products in yields up to 73% with enantiomeric ratio up to 98:2. When electron donating and electron withdrawing groups were employed at 6-position of the coumarin core, the corresponding products (**3b-3g**) were furnished in yields up to 72% and enantiomeric ratio up to 98:2. Additionally, the effect of the electron donating and electron withdrawing groups at the 7- and 8- position of the coumarin core was also evaluated, and the corresponding homologated products (**3h-3n**) were obtained in moderate to high yields up to 70%, and enantiomeric ratios up to 98:2. Next, when both electron donating and electron withdrawing groups were substituted at 6- and 8-positions of the coumarin core, the desired products (**3o-3r**) were obtained in moderate yields and good to excellent enantiocontrol. Moreover, the coumarin core bearing chloro- and methyl- substituents at 6- and 7- positions,

## Table 1 | Optimization Studies

**Organocatalysts**

Ar = 3,5-(CF₃)₂C₆H₃

| Entry | Catalyst | Solvent | Temp. (°C) | Yield (%)[b] | er[c] |
|-------|----------|---------|------------|----------|-------|
| 1 | C1 | DCM | rt | 44 | 76:24 |
| 2 | C2 | DCM | rt | 61 | 47:53 |
| 3 | C3 | DCM | rt | 38 | 34:66 |
| 4 | C4 | DCM | rt | traces | - |
| 5 | C5 | DCM | rt | traces | - |
| 6 | C6 | DCM | rt | 50 | 93:7 |
| 7 | C6 | CHCl₃ | rt | 72 | 95:5 |
| 8 | C6 | DCE | rt | 34 | 91:9 |
| 9 | C6 | THF | rt | traces | - |
| **10** | **C6** | **CHCl₃** | **0** | **73** | **97:3** |

[a]Unless otherwise noted, reaction conditions: **1a** (0.1 mmol), **2a** (0.12 mmol), **Cat** (20 mol%),
[b]Isolated yield,
[c]er calculated by Chiral HPLC.

respectively, yielded **3 s** in 65% yield with 95:5 er. Interestingly, product **3t** containing naphthyl core could be obtained in 71% yield with an excellent enantiomeric ratio of 98:2.

Having established the scope of coumarins, we then moved to investigate the scope of cyclopent-4-ene-1,3-diones (**2b-2ad**) bearing diverse functional groups such as aryl, alkyl, alkenyl, and heterocyclic group. Pleasingly, all the functional groups were well tolerated, thus highlighting the accessibility to a broad library of these desymmetrizing homologating products (Fig. 3). First, variations of the electron-rich substituents (−Me, −OH, and −OMe) at the *para*-position of the aryl ring were examined. The corresponding homologated products (**4a-4c**) were obtained in excellent yields up to 81% and excellent selectivities up to 98:2 er. Gratifyingly, when allyloxy and alkenyloxy groups were introduced at the *para*-position of the aryl ring, provided corresponding **4d** and **4e** in good yields with excellent stereocontrol. Also, to demonstrate the effect of electron-withdrawing groups, various groups such as −F, −Cl, −Br, −CF₃, −CN, and −NO₂ were employed at 4- and 2- positions of the aryl core, which allowed us to access the corresponding homologated products (**4f-4j**) in yields up to 87% with enantiomeric ratio up to 97:3. Additionally, the electronic effect of substituents (−OMe and −Cl) at *meta*-position of the aryl ring, and dihalogen-substituted 1,3-diketones such as 3-bromo-4-chloro, and 4-fluoro-3-methoxy, was also established under the optimal conditions, afforded the corresponding products (**4l-4o**) in moderate to

high yields with good to high selectivities. On the other hand, bulky substitutents such as naphthalen-2-ylmethyl, naphthalen-1-ylmethyl, and benzhydryl underwent the transformation with moderate chemical yields and high selectives (**4p-4r**). Further, *n*-propyl, allyl, cinnamyl, and propargyl substituted substrates were smoothly transformed into expected homologated products (**4s-4v**)in yields up to 87% and up to 92:8 er. Even methyl acetate and thienyl-substituted substrates could furnish the desired products (**4w** and **4x**) in low to good yields with moderate to excellent selectivities. Finally, substrates tethered with biologically relevant scaffolds such as cholic acid, palmitic acid, gemfibrozil, indomethacin, and (S)-naproxen were tolerated exceptionally well with consistently excellent stereocontrol. For instance, the homologated product **4 y** from cholic acid could be furnished in good yield with excellent diastereoselectivity. Importantly, the homologated products (**4z-4ac**) from palmitic acid, gemfibrozil, indomethacin, and naproxen were obtained in high yields and excellent selectivities, which is a testament to the robustness of our methodology. Moreover, some of the coumarins (**1j, 1o,** and **1 s**) and cyclopent-4-ene-1,3-diketones (**2b, 2j, 2n,** and **2p**) substrates were also tested under the slightly modified catalytic system. The homologating annulation reaction with **1j, 2b,** and **2n** by utilizing 10 mol% of **C6**, at 0 °C for 3 d, and afforded the corresponding products comparatively in lower yields with similar selectivities. While substrates **1o, 1 s, 2j,** and **2p** when screened under the same conditions yielded the desired products in similar yields with slightly enhanced selectivities (Supplementary Table 4 for details).

The absolute configuration of **3a** was established by the X-ray diffraction analysis of its single crystal obtained from an *n*-heptane/CHCl₃ mixture and was found to be *R* (Fig. 2). The absolute configurations of the other homologated products were assigned by analogy as the same.

A plausible mechanistic pathway for our protocol has been demonstrated in Fig. 4. The domino homologating annulation reaction is believed to proceed through Michael addition forming intermediate **I**, following Thrope-type reaction that results in the genesis of intermediate **II**. The intermediate **II** thus formed undergoes intramolecular aldol reaction leading to the generation of intermediate **III**, which further takes part in semi-pinacol type ring expansion, and subsequent intramolecular Mannich reaction/ring contraction giving intermediate **IV**. Further, the intermediate **IV** involves cyclopropane ring opening event to form intermediate **V** that provides the desired homologating annulated products **3a** through isomerization of double bonds followed by silica gel promoted oxidation. Additionally, when we monitored the reaction progress through ¹H NMR, it was revealed that the peak at 10.29 ppm does not correspond to the desired product **3a**, instead corresponds to NH of the intermediate **V** (Fig. 4C). Surprisingly, after 72 h of reaction time, when silica gel was introduced to the reaction mixture, the deep brown reaction mixture changes to deep blue which eventually marks the appearance of characteristic peak of the desired product **3a** at 9.8 ppm (Fig. 1B. Further, the ESI-MS analysis of the reaction mixture also supports the formation of intermediate **V** during the reaction progress as shown in Fig. 4C (Supplementary Fig. 1A–C for details).

Further based on the reaction outcome and previous literature reports we propose a model for the origin of enantioselectivity. The γ-C-H of 3-cyano-4-methylcoumarin **1a** gets deprotonated by the tertiary amine moiety of the catalyst **C6** and the resulting dienolate gets stabilized by the two thiourea NH groups. Subsequently, the electrophile **2a** (2-benzyl-2-methylcyclopent-4-ene-1,3-dione) gets hydrogen bonded with the protonated pyrrolidine moiety. To sustain these robust stabilizing interactions in the two transition states (**TS-I** and **TS-II**), the switching of substituents (Me and Bn) at the quaternary chiral carbon center of 1,3-diketone is necessary. As a result, a notable distinction between the two transition states arises, and a stabilizing electrostatic

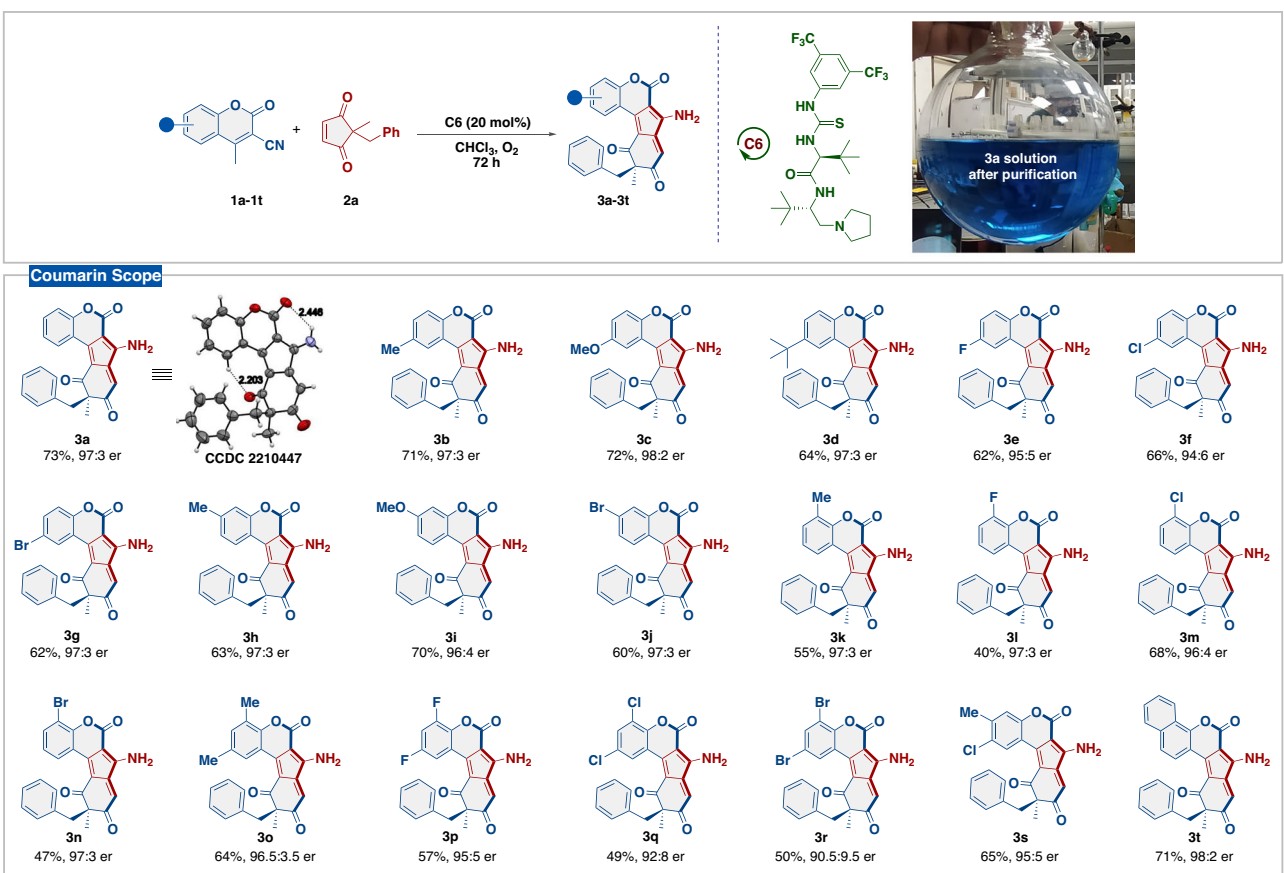

**Fig. 2 | Coumarin Scope**[a–c]. [a]Unless otherwise noted, reaction conditions: **1** (0.1 mmol), **2a** (0.12 mmol), **C6** (20 mol%), in CHCl₃ (1 mL) and O₂ as an oxidant. [b]Isolated yield, [c]er calculated by Chiral HPLC.

interaction occurs exclusively in the **TS-I** leading to the formation of major enantiomer (*R*)-**3a** involving an aromatic CH in the benzyl group of 1,3-diketone and one of the CF₃ groups in the catalyst (**C6**) (Fig. 4B)[26–28].

After establishing the scope of our protocol, its synthetic utility was manifested by scaling up the model reaction to a 6 mmol scale. For this transformation, the catalyst **C6** loading was reduced to 10 mol%, and reaction time was increased to 4 d. The homologated product **3a** was obtained in a slightly lower yield (68%) with similar selectivity as established in the optimizing stage. After isolating **3a** through column chromatography the catalyst **C6** was also eluted. The catalyst was then recrystallized with a CHCl₃/n-pentane solvent system and subsequently subjected to the model reaction under the optimal reaction conditions, which again afforded **3a** in similar yield and selectivity as established in the optimization stage, thus showing the robustness of our catalytic system (Fig. 5).

Next, to show the synthetic utility of the developed compounds, **3a** was treated with 3, 5-bis(trifluoromethyl)phenyl isocyanate, and 3, 5-bis(trifluoromethyl)benzoyl chloride, the corresponding thiourea and amide **5** and **6**, could be obtained in high yields with excellent stereocontrol. The bromo analog **7** of homologated adduct **3a** could easily be afforded by converting −NH₂ group on treatment with NaNO₂ and HBr in 50% yield with 97:3 er. Further, on heating at 180 °C for 16 h, **3a** was successfully converted into corresponding 8-benzylated derivative **8** in 65% yield and 94:6 er. Finally, the reaction of **3a** with NBS yielded the corresponding brominated adduct **9** in excellent yield with excellent stereo retention.

After establishing the scope and synthetic applicability of our strategy, we then proceeded to investigate crystal packing, photophysical, and biological properties of these highly substituted

polycyclic motifs containing embedded aminopentafulvene core. Analyzing the crystal packing of compound **3a**, revealed a distinctive A-B-A-B pattern along the *b-c* plane as seen in Fig. 6A. Along the *b-c* plane (Fig. 6B), the crystal packing pattern showed two types of strong hydrogen bondings among the molecules in A-B pairs, i.e., N-H···O (2.24 Å) and alkenic C-H···O (2.39 Å). Additionally, another hydrogen bonding interaction between aromatic C-H···O (2.49 Å) was noticed among the parallely arranged molecules in the *b-c* plane (A-A grey colored and B-B pink colored). Moreover, when the packing pattern was analyzed along the *a-b* plane (Fig. 6C), an interesting three-point interaction, i.e., O···C-N (3.19 Å), and two O···C-C (3.01 Å and 3.02 Å) were identified. Furthermore, the homologated product **3a** also showed two additional types of intramolecular interactions, i.e., N-H···O (2.45 Å) and a strong Ar C-H···O (2.20 Å) (Fig. 6D). Interestingly the effect of C-H···O interaction was translated in the ¹H NMR spectra of the homologated products by deshielding the corresponding H atom to 9.8 ppm. These interesting packing patterns suggest the potential applications of the developed homologated analogs as electrochromic material and in solar cell development.

Further, as the homologated products integrate two photoactive cores with known emission characteristics. The fulvene core and the coumarin core, we anticipated that these adducts might possess interesting photophysical properties. Thus, we investigated the emission characteristics of homologated products with electron donating and electron withdrawing substituent modifications on the benzo core of coumarin moiety (3 Series) and differently substituted 1,3-diketone core (4 series). On UV excitation λₑₓ = 360 nm for **3a** and all other molecules, we observed emission peaks at ~440 nm and ~575 nm. Interestingly, the peaks were more dominant and distinct in the chiral molecules compared to their racemic analogs, again highlighting the

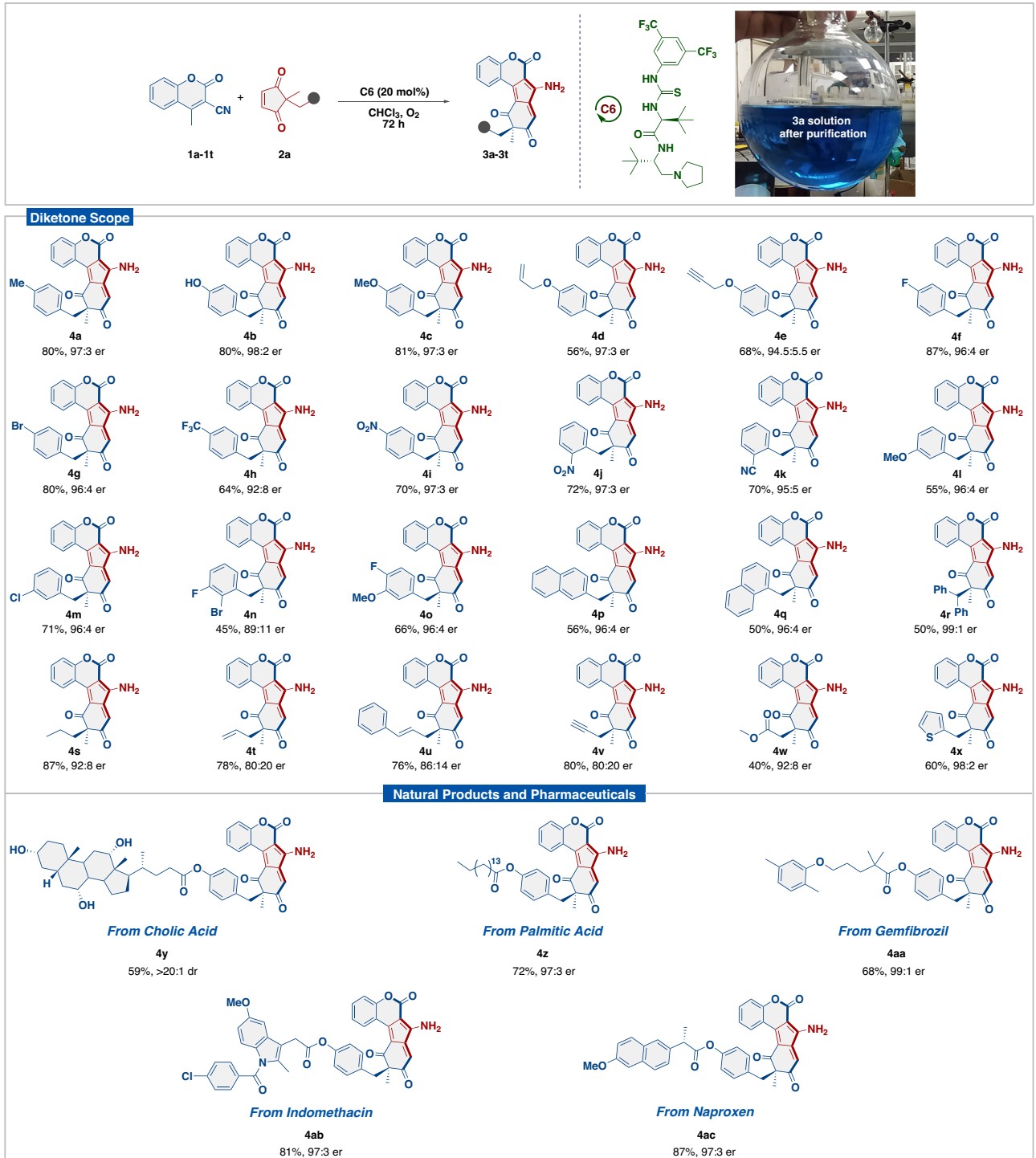

**Fig. 3 | Scope of diketone**[a-c]. [a]Unless otherwise noted, reaction conditions: **1a** (0.1 mmol), **2** (0.12 mmol), **C6** (20 mol%), in CHCl₃ (1 mL) and O₂ as an oxidant. [b]Isolated yield, [c]er calculated by Chiral HPLC.

importance of chirality in the developed compounds (Supplementary Fig. 2a for details). We were delighted to observe that some chiral molecules such as **3c**, **3p**, and **3t** had excellent emission peaks (Fig. 7a). This also indicated that probably the coumarin cores were influencing the fluorescence properties more as various substituents on the 1,3-diketone core of aminopentafulvene did not affect the emission peak intensity or the wavelength. As expected, the delocalization of the electrons on the coumarins by extending the conjugation via substitution of methoxy, fluorine or extented benzene ring, as in **3c**, **3p** and **3t** respectively, resulted in increased emission intensity.

Encouraged by the emission spectra, we explored if these near-red fluorescent dyes can be used for bioimaging. We first determined the cytotoxicity of the molecules by the standard tetrazolium bromide (MTT) assay, which spectrophotometrically measures the mitochondrial activity of mammalian cells (Supplementary Fig. 2b for details). All the molecules were found to be well tolerated at a concentration of 10 μM. We also confirmed the viability of cells by the live-dead assay using the live cell fluorescent dye, calcein acetoxymethyl ester (Calcein-AM), and the dead cell dye propidium iodide. As shown in Fig. 7b, all the compounds including the control showed more than 90% cell

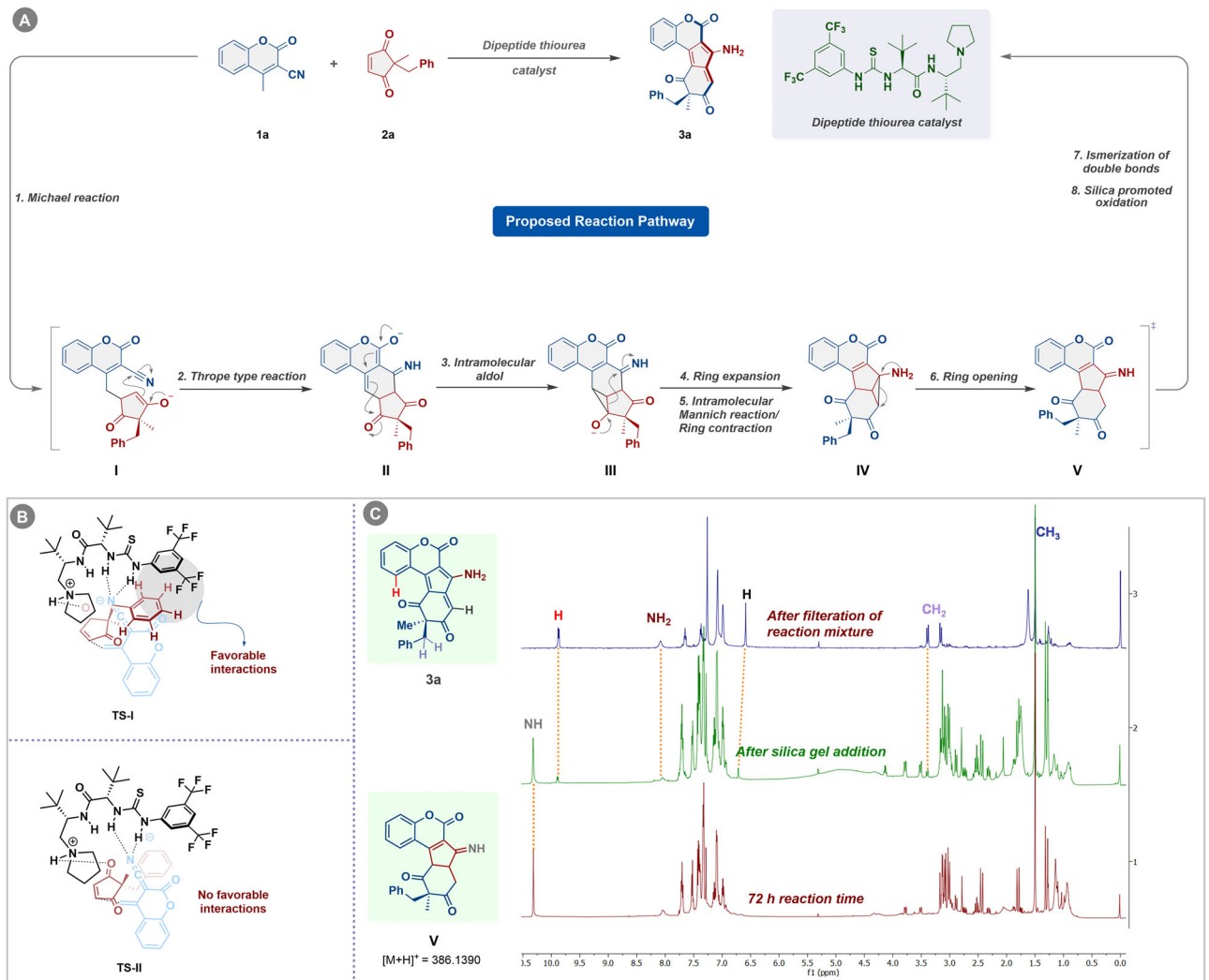

**Fig. 4 | Overview of catalytic cycle, origin of enantioselectivity and mechanistic insights. A** Proposed reaction pathway. **B** The proposed transition **TS-1** depicts the favorable interactions between the aromatic CH of benzyl group and one of the CF₃ of the catalyst. The **TS-II** suggests no such favorable interactions responsible for the origin of enantioselectivity. **C** Reaction progress monitoring through ¹H NMR.

viability after 24 h of incubation, and there was no significant difference between the compounds and the control group. The results indicated no cytotoxicity of the developed compounds towards the healthy mammalian fibroblasts (NIH-3T3) cells. Furthermore, the live–dead assay was also consistent with the MTT results, as all the compounds showed a high number of live cells, as seen in Fig. 7c. After confirming the cytocompatibility of the developed compounds, we evaluated the potential application of the compounds in bioimaging. We incubated NIH-3T3 cells with the developed compounds and observed the fluorescence after washing the cells. As can be seen from fluorescence microscopy images (Fig. 7d), all compounds were easily internalized into the cells after 4 h of incubation. A strong red fluorescence from the cell cytoplasm was observed for cells incubated with compounds, especially for molecule **3t**. Thus, the cytotoxicity data and uptake experiment together indicate that the developed compounds can be used for intracellular imaging without adverse cytotoxic effects and can be explored further for such applications.

In summary, we have unraveled an L-*tert*-leucine derived thiourea catalyzed enantioselective homologating annulation of cyclopent-4-ene-dione with 3-cyano-4-methylcoumarins giving access to a wide range of enantioriched polycyclic multi-substituted amino pentafulvene cores. Importantly, our catalytic system efficiently catalyzed this single-step transformation

involving substrates tethered with natural products and drug candidates, providing the complex homologated adducts in high yields with excellent stereocontrol. The cytotoxicity and cellular uptake experiments revealed that the enantiopurity of this class of polycyclic aminopentafulvenes has significant effects on their photophysical properties and cell viability. We believe that our discovery will not only offer these stereochemically diverse motifs but will be a guiding light in the field of organocatalytic homologations and multistep synthesis.

## Methods
### General procedure for asymmetric vinylogous homologating annulation
In an oven and vacuum-dried reaction tube, catalyst **C6** (11.1 mg, 0.02 mmol, 0.2 equiv), and 1,3-diketone **2** (0.12 mmol, 1.2 equiv.) were taken in freshly CHCl₃ at room temperature. The reaction mixture was cooled at 0 °C. Subsequently, 3-cynao-4-methylcoumarin **1** (0.1 mmol, 1.0 equiv.) was charged to the reaction mixture in one shot under open air/O₂ atmosphere. The resulting mixture was allowed to stir vigorously at the same temperature for 72 h and the progress of the reaction was monitored through TLC. Once the reaction was completed, the reaction mixture was quenched with silica-gel 200 mg and was further stirred at rt for 30 mins. The reaction was then directly processed for

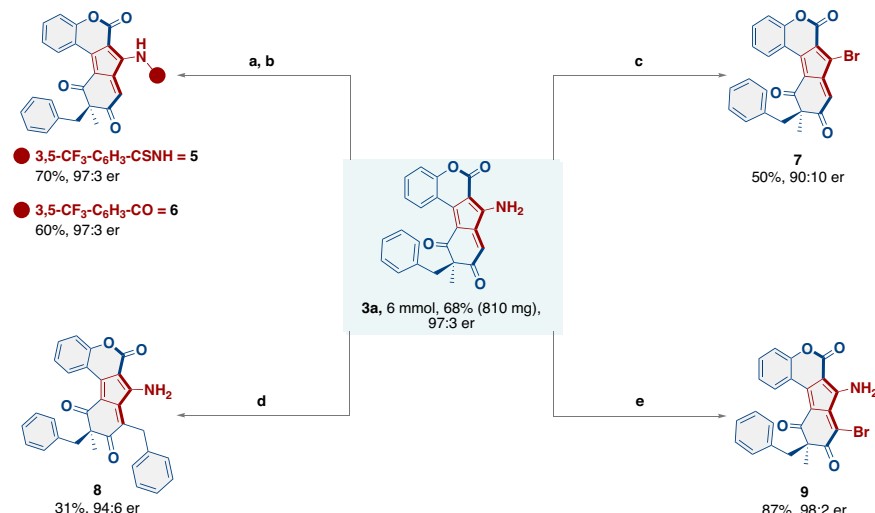

**Fig. 5 | Scale up experiment for 3a and its synthetic transformations. a 3a** (0.1 mmol, 1 equiv.), 3,5-bis(trifluoromethyl)phenyl isocyanate (0.1 mmol, 1 equiv.), DCM (1 mL), rt, 8 h. **b 3a** (0.1 mmol, 1 equiv.), 3,5- bis(trifluoromethyl)benzoyl chloride (0.12 mmol, 1.2 equiv.), DCM (1 mL), rt, 12 h. **c 3a** (0.1 mmol 1 equiv.),

NaNO₂, 0.15, 48 % HBr 250 μL, −50 °C 1 h, then add 2 mL of Et₂O, −8 °C for 2 h, again cooled to −40 °C, 100 mg Na₂CO₃, stirred overnight. **d 3a** (0.1 mmol, 1 equiv.) in toluene 1 mL, heated at 180 °C, 16 h. **e 3a** (0.1 mmol, 1 equiv., NBS (0.5 mmol, 5 equiv.). DMF, rt, 1 min.

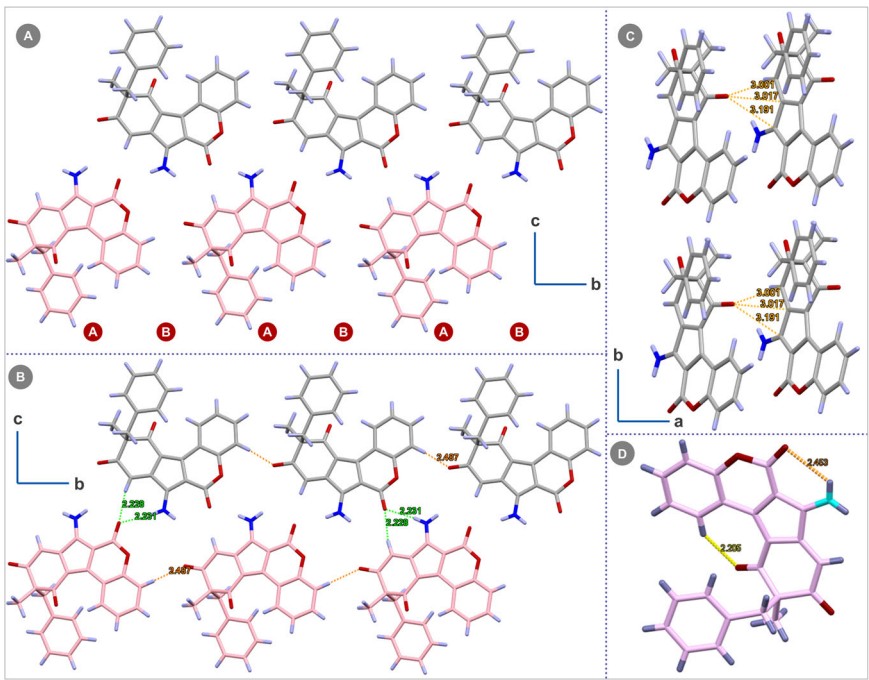

**Fig. 6 | Analysis of crystal packing and molecular interactions in 3a. A** A-B-A-B crystal packing pattern in *b-c* plane for **3a**. **B** The *b-c* plane shows Alkene C-H---O, Ar C-H---O, and N-H---O interactions in **3a**. **C** The a-b plane depicting HN-C---O, 2 alkene C = C---O interactions in **3a**. **D** Showing intramolecular H-Bonding.

purification by silica gel column chromatography without any workup to afford chiral products **3** and **4**.

**Note:** After 72 h of reaction time, we observed that the reaction mixture becomes brownish in color. When 200 mg silica gel was added to the reaction mixture, the color of the mixture turned deep blue, which is the color of our products.

### Absorption and fluorescence spectroscopy

The absorption and fluorescence spectra of the compounds (**3c, 3p,** and **3t**) were measured using a BioTek Synergy H1 Multimode reader. The absorbance spectra of the compounds were recorded at

wavelengths ranging from 300 to 700 nm. Further, the aqueous solution of compounds was excited at a particular wavelength (360 nm as observed by absorbance spectra) to obtain an emission spectra.

### Cytotoxicity

The cytotoxicity of the developed compounds was measured using 3-(4, 5-dimethylthiazol-2-yl)-2, 5-diphenyl tetrazolium bromide (MTT, Thermo Fisher) and live-dead assay. Cells were seeded on to the glass coverslips at a density of 10,000 cells/well and were allowed to adhere for 24 h. 1% DMSO and tissue culture plate were taken as control. After

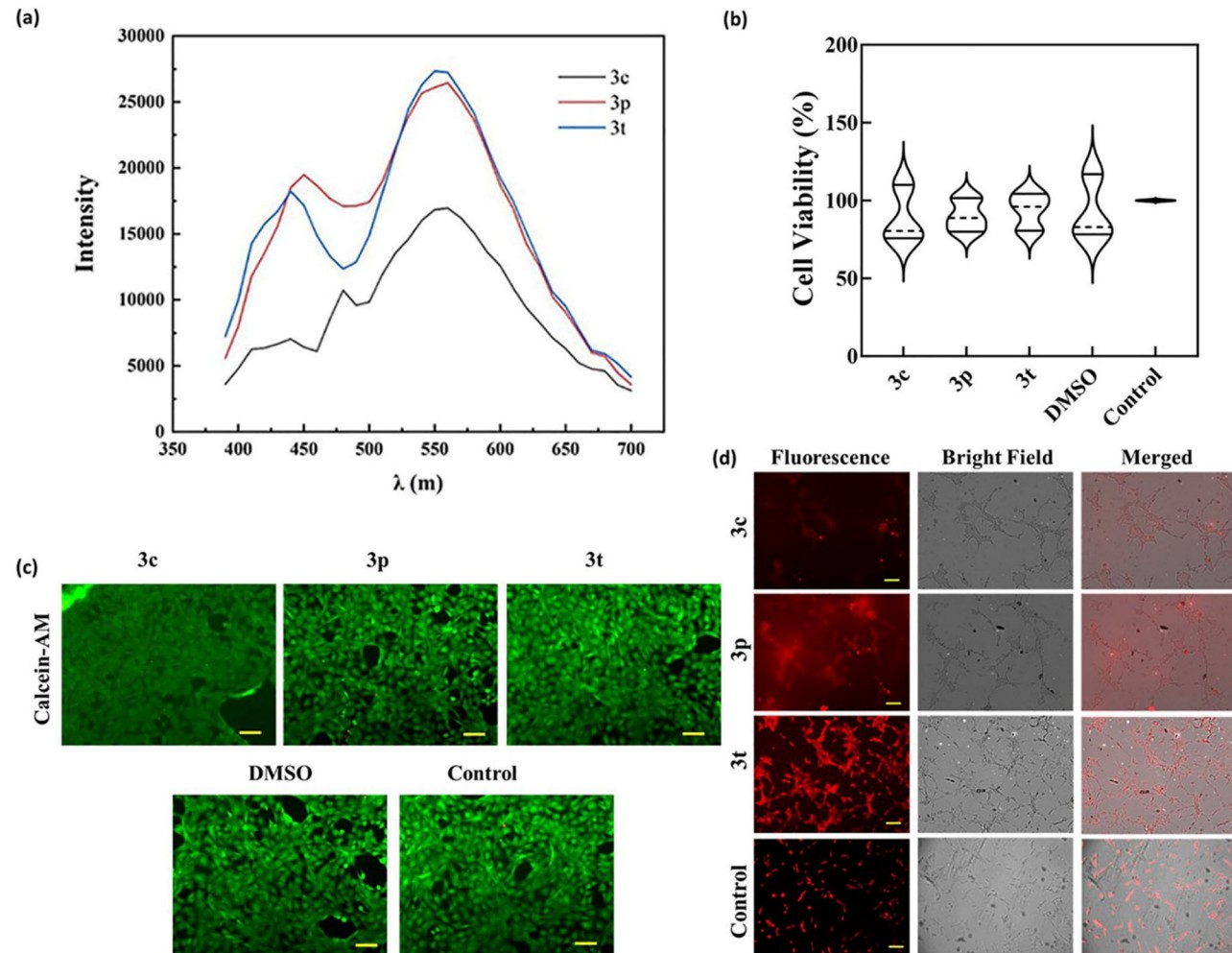

**Fig. 7 | Photophysical, cytocompatibility, and cellular uptake investigations.** **a** Excitation-dependent fluorescence spectra of the developed compound, **b** MTT and in-vitro cytocompatibility of the developed compounds **c** live-dead assay for 24 h, and **d** cellular labeling of NIH-3T3 cells with the developed compounds. (Scale bar: 100 μm). The data represents the mean and standard deviation of three independent samples ($n = 3$).

24 h, the media was replaced with fresh media containing 1% compound and allowed to grow for another 24 h and were then stained with calcein-AM (Thermo Fisher) and propidium iodide (Thermo Fisher) and imaged under a fluorescence microscope. For MTT assay, the MTT reagent was added for 4 h followed by addition of dimethyl sulfoxide and OD measurement at 570 nm.

## Cellular uptake by microscopy

Cells (NIH-3T3, purchased from National Center for Cell Sciences (a National Institute with cell repository in India), Pune, India,) were cultured in high glucose Dulbecco's modified Eagle's medium (DMEM, GIBCO) with 10% fetal bovine serum (FBS, GIBCO) under 5% $CO_2$ at 37 °C until use. Cover slips (circular) were first sterilized using UV and ethanol treatment and then were washed 3 times with PBS before use. Cells at a density of 10,000 cells/well were seeded onto the coverslips and were allowed to grow for 24 h. After 24 h of culture, the media was replaced with fresh media containing 1% of ($10^{-3}$ molar) compound and was incubated for 4 h. After 4 h, the media was removed and washed 3 times with PBS and was imaged on a fluorescence microscope using Alexa flour 594 filter.

## Reporting summary

Further information on research design is available in the Nature Portfolio Reporting Summary linked to this article.

## Data availability

Materials and methods, experimental procedures, useful information, mechanistic studies, [1]H NMR spectra, [13]C NMR spectra, HPLC chromatograms, and mass spectrometry data are available in Supplementary Information. Raw data is available from the corresponding author upon request. Crystallographic data for compound **3a** has been deposited with the Cambridge Crystallographic Data Centre under the accession number CCDC 2210447. These data are provided free of charge from The Cambridge Crystallographic Data Centre via www. ccdc.cam.ac.uk/structures.

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

## Acknowledgements

We are grateful for the generous financial support from SERB-INDIA(CRG/2021/006502) and SERB (SCP/2022/000599). SS and RS are grateful to the UGC and CSIR, New Delhi, India, for SRF and AJ to MHRD for financial assistance. We thank DST–FIST for the mass spectrometer facility and CRF–IITD for providing NMR facilities.

## Author contributions

R.P.S. and S.S. conceived the project and designed the experiments. S.S. and R.S. performed the experiments and analyzed the data with guidance from R.P.S. A.J. conducted the biological experiments. N.S. mentored A.J. and S.S. in perofoming the biological studies. R.P.S., S.S., R.S. N.S. and A.J. prepared this manuscript. All authors discussed the results and contributed to the final paper.

## Competing interests

The authors declare no competing interests.
