## [Peer Review File · Nature Communications]

Desymmetric Homologating Annulation to Access Chiral Pentafulvenes and Their Application in BioimagingREVIEWER COMMENTS

Reviewer #1 (Remarks to the Author):

Polycyclic compounds have important applications in modern organic synthesis. In this manuscript, Singh and coworkers reported a chiral thiourea catalyzed enantioselective homologating annulation of cyclopent-4-ene-dione with 3-cyano-4-methylcoumarins, providing a wide range of enantioriched polycyclic multisubstituted amino pentafulvene cores. In the tandem reaction, an unusual asymmetric desymmetric homologating annulation process was involved to provide enantioenriched pentafulvenes. Furthermore, cytotoxicity and cellular uptake experiments were conducted to show the potential utility of the cyclization products. The paper is recommended for publication in Nature Communications after certain revisions are implemented, as indicated below:

1. The loading of catalyst is too high (20 mol%), although 10 mol% loading was used in the 6.0 mmol scale experiment.
2. The yields of the products were not good enough (generally 50-60%).
3. The proposed reaction pathway was too simple, the critical stereoselective control model was not displayed.
4. Scheme 2, compared to compound 3a, the ee value of compounds 5 and 6 had an obvious improvement. The authors should give a reasonable explanation.
5. The NMR monitor experiment was rough, and no valuable information was disclosed.
6. In the Supporting Information, the specific rotation for the enantioenriched compounds should be given.

Reviewer #2 (Remarks to the Author):

The authors report that a L-tert-leucine derived thiourea catalyzed enantioselective homologating annulation of cyclopent-4-enedione with 3-cyano-4-methylcoumarins giving access to a various enantioriched polycyclic multisubstituted amino pentafulvene cores. The reaction itself is attractive and has broad utility to give the amino pentafulvenes. The reaction mechanism is also reasonable. However, the authors do NOT provide the rational explanation of the origin of enantioselectivity in the manuscript or even in the Supporting Information. This point is a fatal defect of this manuscript. Additionally, there are several errors in the manuscript. One is a typo in page 8 line 8, "ulky substituted". The other is that the authors do NOT follow the rule of International System (SI) of Units in the caption of Scheme 2 and the Supporting Information. For all of these reasons, the referee will not recommend that the manuscript is accepted for publishing in this journal.

Response to Reviewer

Reviewer: 1

Comments to the Author

1) Comment: *Polycyclic compounds have important applications in modern organic synthesis. In this manuscript, Singh and coworkers reported a chiral thiourea catalyzed enantioselective homologating annulation of cyclopent-4-ene-dione with 3-cyano-4-methylcoumarins, providing a wide range of enantioriched polycyclic multisubstituted amino pentafulvene cores. In the tandem reaction, an unusual asymmetric desymmetric homologating annulation process was involved to provide enantioenriched pentafulvenes. Furthermore, cytotoxicity and cellular uptake experiments were conducted to show the potential utility of the cyclization products. The paper is recommended for publication in Nature Communications after certain revisions are implemented, as indicated below.*

Response: Thank you for considering our work.

2) Comment: *The loading of catalyst is too high (20 mol%), although 10 mol% loading was used in the 6.0 mmol scale experiment*

Response: We acknowledge the reviewer's concern regarding alleged high catalyst loading. Although we have already performed the scale up reaction for the model substrate at 10 mol% catalyst loading, a slight decrease in reactivity was observed as displayed in the manuscript. However, to address the reviewer's concern, we have executed a few more entries with 10 mol% catalyst loading for 3 and 6 days respectively. It is revealed that the reactivity is very slow at 10 mol% catalyst loading however the reactions that were allowed to stir for 6 days showed a similar level of reactivity and a slightly enhanced selectivity as compared to the

results obtained at 20 mol% catalyst loading. Therefore, to reduce the reaction completion time we have utilized 20 mol% catalyst loading.

3) Comment: *The yields of the products were not good enough (generally 50-60%).*

Response: We appreciate the reviewer's concern regarding the yields of some of the products. As we know the steric and electronic effects of the different substituents impact the reactivity and selectivity of a reaction outcome. Here, in our protocol, we have kept the constant reaction time. Therefore, due to the difference in the reactivity of the different substrates, some of the substrates have been obtained with lower yields. However, if you increase the reaction time there is a definite increase in the yields of the products without impacting the stereo outcome of the reaction as established in the above response (Comment 2).

4) Comment: *The proposed reaction pathway was too simple, the critical stereoselective control model was not displayed.*

Response: We thank the reviewer for bringing this to our notice. In the revised manuscript, we have included the model for the origin of the enantioselectivity. Based on previous reports and reaction outcomes, we propose that the electrophile (2-benzyl-2-methylcyclopent-4-ene-1,3-dione) gets hydrogen bonded with the protonated pyrrolidine moiety, while the two thiourea NH groups play a role in stabilizing the dienolate (3-cyano-4-methyl-coumarin) through nitrogen atom, the switching of substituents (Me and Bn) at the quaternary chiral carbon center is necessary to sustain these robust stabilizing interactions in the two transition states (TS-I and TS-II). As a result, a notable distinction between the two transition states arises, and a stabilizing electrostatic interaction occurs solely in the TS-I leading to the predominant product enantiomer (**R**)-**3a** involving an aromatic CH in the benzyl group and one of the CF₃ groups in the catalyst.

References:

1. Chandra Mallojjala, S.; Sarkar, R.; Karugu, R. W.; Manna, M. S.; Ray, S.; Mukherjee, S.; Hirschi, J. S. *J. Am. Chem. Soc.* **2022**, *144* (38), 17399–17406.
2. Singh, S.; Sunoj, R. B. *Advances in Physical Organic Chemistry*; Williams, I. H., Williams, N. H., Eds.; Academic Press, **2019**; Vol. 53, Chapter 1, pp 1–27.
3. Bhaskararao, B.; Sunoj, R. B. *Chem. Sci.* **2018**, *9*, 8738–8747.

5) Comment: *Scheme 2, compared to compound 3a, the ee value of compounds 5 and 6 had an obvious improvement. The authors should give a reasonable explanation.*

Response: We thank the reviewer for bringing this to our notice. As can be seen in ESI, the chromatograms corresponding to compounds 5 and 6, one of the peaks of the two enantiomers of both compounds 5, and 6 are highly trailing. However we have rectified the chromatograms in the revised supporting information.

6) Comment: *The NMR monitor experiment was rough, and no valuable information was disclosed.*

Response: Thank you for your valuable observation. Indeed we have conducted the NMR experiment for monitoring the reaction progress. As can be seen in Manuscript (Fig. 2B) and ESI (Fig. S1A and S1B), the limiting reagent **1a** almost gets consumed after 72 h of reaction time and there are no characteristic peaks seen in ¹H NMR spectrum with respect to the homologated product instead the peaks correspond to the intermediate **V** (also confirmed by HRMS). It is only when the silica gel is added to the reaction mixture, that the intermediate **V** starts converting into the product which is evident in the ¹H NMR spectrum (Fig 2B, S1A, and S1B). However, we have made some necessary changes in the manuscript to demonstrate clear information.

7) Comment: *In the Supporting Information, the specific rotation for the enantioenriched compounds should be given.*

Response: We acknowledge the reviewer's concern regarding the inclusion of specific rotation of the enantioenriched compounds. Primarily we attempted to record the optical rotation of the developed compounds using a 100 mm polarimeter cell, but due to the highly dark solution of the compounds we did not get success. However, in the revised ESI, we have now included the specific rotation in supporting information using a 10 mm polarimeter cell.

Reviewer: 2

Comments to the Author

The authors report that a L-tert-leucine derived thiourea catalyzed enantioselective homologating annulation of cyclopent-4-enedione with 3-cyano-4-methylcoumarins giving access to a various enantioriched polycyclic multisubstituted amino pentafulvene cores. The reaction itself is attractive and has broad utility to give the amino pentafulvenes. The reaction mechanism is also reasonable. However, the authors do NOT provide a rational explanation of the origin of enantioselectivity in the manuscript or even in the Supporting Information. This point is a fatal defect of this manuscript. Additionally, there are several errors in the manuscript. One is a typo in page 8 line 8, "ulky substituted". The other is that the authors do not follow the rule of International System (SI) of Units in the caption of Scheme 2 and the Supporting Information. For all of these reasons, the referee will not recommend that the manuscript is accepted for publishing in this journal.

Response: We appreciate the reviewer's valuable comments and concerns. We have addressed every concern raised by the reviewer. We have included the probable transition state responsible for the origin of the enantioselectivity based on the previous reports and reaction outcomes. The reviewer's concern regarding the typos and other technical errors in the manuscript have been addressed and necessary changes have been made in the revised manuscript and supporting information.

REVIEWERS' COMMENTS

Reviewer #1 (Remarks to the Author):

After carefully evaluating the revised manuscript, including the response to the catalyst loading, the relatively low yields of the products, the proposed control model, the NMR experiments, and even the specific rotation of the products, the reviewer thought that this manuscript did not meet the high standard of the Nature Communications. Publication in a more specific journal, such as Communications Chemistry is recommended.

Reviewer #2 (Remarks to the Author):

The authors reviewed previous manuscript, and the authors provided the rational explanation of the origin of enantioselectivity in the manuscript. The proper references are cited to support the explanation. Now the manuscript is well matured and reasonable. Therefore, the referee recommends that the manuscript is accepted for publishing in Nature Communications after the authors review and consider following points.

- (1) The authors did not give any explanations of Figure 1 D and E. Please mention them in the manuscript.
- (2) (4+2) annulation can be changed to [4+2] annulation, however, both expressions can be found in papers. So, it depends on the authors.
- (3) Do not miss the period just after "Fig". For example, in lines 66, 68, 70, and 71, the authors wrote "Fig 1B(i)", "Fig 1B(ii)", "Fig 1B(iii)", and "Fig 1C". They should be changed to "Fig. 1B(i)", "Fig. 1B(ii)", "Fig. 1B(iii)", and "Fig. 1C".
- (4) Please clean up the Figure 3.
 - (a) In TS-I, the minus charge should be put near the nitrogen atom originated by the cyano group. Please do not omit the carbon atom of C-C-N double-double bonds originated by the cyano group. Now it is not shown, and C-C-N double-double bond looks like a long C-N double bound.
 - (b) The referee recommend that the authors show the H atoms on the aromatic ring of the benzyl group, because the authors pointed out the interaction between "an aromatic CH in the benzyl group and one of the CF₃ groups in the catalyst".
- (5) In line 275, "0°C" should be "0 °C". Insert the space between the figure and unit. Please check the manuscript and Supporting Information carefully again.

Response to Reviewer

Reviewer: 1

Comments to the Author

1) Comment: *After carefully evaluating the revised manuscript, including the response to the catalyst loading, the relatively low yields of the products, the proposed control model, the NMR experiments, and even the specific rotation of the products, the reviewer thought that this manuscript did not meet the high standard of the Nature Communications. Publication in a more specific journal, such as Communications Chemistry is recommended*

Response: The publication of our work in a prestigious journal like Nature Communications is critical for its visibility. This work will be quite beneficial to a lot of researchers in many cross-disciplinary areas. The broad readership of Nature Communications is apt for this rather than narrow readership of a field specific journal.

Reviewer: 2

Comments to the Author

The authors reviewed previous manuscript, and the authors provided the rational explanation of the origin of enantioselectivity in the manuscript. The proper references are cited to support the explanation. Now the manuscript is well matured and reasonable. Therefore, the referee recommends that the manuscript is accepted for publishing in Nature Communications after the authors review and consider following points.

Response: We thank the reviewer for considering our manuscript.

1) Comment: *The authors did not give any explanations of Figure 1 D and E. Please mention them in the manuscript.*

Response: We thank the reviewer for pointing out this inadvertent mistake. We have now mentioned the explanation for Figures 1D and 1E in the main text.

2) Comment: *(4+2) annulation can be changed to [4+2] annulation, however, both expressions can be found in papers. So, it depends on the authors.*

Response: We thank the reviewer for the suggestion. Although both representations are being used lately, however, [4+2] notation is used for classical cycloaddition reaction. Our protocol is not the case of classical cycloaddition reaction hence we stick with the (4+2) annulation expression.

3) Comment: *Do not miss the period just after “Fig”. For example, in lines 66, 68, 70, and 71, the authors wrote “Fig 1B(i)”, “Fig 1B(ii)”, “Fig 1B(iii)”, and “Fig 1C”. They should be changed to “Fig. 1B(i)”, “Fig. 1B(ii)”, “Fig. 1B(iii)”, and “Fig. 1C”.*

Response: We appreciate the reviewer for identifying these typos. We have rectified them in the revised manuscript.

4) Comment: *(4) Please clean up the Figure 3.*

Response: We thank the reviewer for the suggestion. We have cleaned up Figure 3 (after the editorial team's suggestion **Figure 3** is now changed to **Figure 6**).

4a) Comment: *In TS-1, the minus charge should be put near the nitrogen atom originated by the cyano group. Please do not omit the carbon atom of C-C-N double-double bonds originated by the cyano group. Now it is not shown, and C-C-N double-double bond looks like a long C-N double bound.*

Response: We thank the reviewer for their valuable comment. We have made the necessary changes as suggested by the reviewer in the revised manuscript.

4b) Comment: *The referee recommend that the authors show the H atoms on the aromatic ring of the benzyl group, because the authors pointed out the interaction between “an aromatic CH in the benzyl group and one of the CF3 groups in the catalyst”*

Response: We thank the reviewer for the suggestion. In the revised manuscript, we have shown the H atoms on the aromatic of the benzyl group in TS-1.

5) Comment: *In line 275, “0°C” should be “0 °C”. Insert the space between the figure and unit. Please check the manuscript and Supporting Information carefully again.*

Response: In the revised manuscript, we have rectified all the typos both in the main manuscript and supplementary information.